# Mental health and lived experience: The value of lived experience expertise in global mental health

Claudia Sartor [ID]

Mental Health, The Global Mental Health Peer Network, Cape Town, South Africa

## Perspective

global mental health; human rights; lived experience; mental health; peer support; quality of life

**Corresponding author:**
Claudia Sartor;
Email: claudia.sartor@gmhpn.org

## Abstract

There is no disputing the current established global consensus that people with lived experience of a mental health condition ("people with lived experience") play an integral role in influencing policy and processes in global mental health. Specifically, the role they hold as agents of change through which they can lead and co-lead projects on mental health, alongside a multidisciplinary team, as recommended in the findings of the report of Lancet Commission on ending stigma and discrimination (Thornicroft et al. [2022], "The Lancet Commission on ending stigma and discrimination in mental health", *Lancet*, 400, 1438–1480). Immense value is associated with their unique expertise not learned through theoretical concept but based on real-life experience. Appreciating their involvement in processes is a human right, supported by international human rights instruments such as the United Nations Convention on the Rights of Persons with Psychosocial Disabilities (2006). However, there remains an expectation that people with lived experience are expected to be involved in processes and service delivery without receiving remuneration for their expertise. This article will provide the basis for which processes must follow the principle of equity; that lived experience expertise ought to be equally compensated for based on equal pay for equal work. In closing, it will provide a recommendation for stakeholders on how to improve upon effective engagement with people with lived experience, leading to meaningful and authentic contributions.

## Impact statement

Emphasising the importance of integrating lived experience into all domains in the mental health sector will enhance further action in a manner that respects and protects human rights. By ensuring that policies and mental health care truly aligns with the needs of lived experience, will enhance suitable conditions for improved mental health and well-being of all people. This article is written as a continued effort to advocate for changes to current global mental health practices. It will, no doubt, strengthen the working relationship between mental health professionals and non-mental health professionals who work alongside people with lived experience. Only once the recommendations are adopted, will global mental health serve people with lived experience and also build a system based on a recovery and person-centred model, within a human rights framework. Through inclusion of diverse lived experience, we will overcome inequalities, inequities and be better equipped to serve peoples' needs based on their local and contextual needs.

## Introduction

This article places particular focus on people with lived experience and the role their expertise brings to discussions on mental health, highlighting their valuable and highly resourceful contributions to the global mental health sector. There are reputable resources corroborating the positive impact and experiential value that lived expertise holds in improving mental health awareness, advocacy, policy review and global best practice. In recent times, it has gained huge recognition and validation from global partners. Despite the current consensus on the experiential value of lived experience expertise, there is a call for betterment in remunerating people with lived experience for actively partaking in decision-making and research projects, lived experience engagement sessions and service provision. This article will provide actionable recommendation on how to bridge the gap of the existing challenges.

## International resources supporting lived experience value

The experiential value that people with lived experience bring to the fore in the global mental health sector is indisputable and supported by two well-established resources of the World Health

Organization (WHO) and its report on "Transforming mental health for all" (WMHR) launched in June 2022 and "The Lancet Commission on ending stigma and discrimination in mental health". The latter launched on 10 October 2022; on World Mental Health Day. It is well noting that both reports developed as a result of years of strong advocacy efforts by people with lived experience, their families and other key role players to achieve fair practices in global mental health and a betterment in law and policy reform.

### The WHO comprehensive mental health action plan 2013–2030

In May of 2013, health ministers from 194 member states joined the 66th World Health Assembly and adopted the WHO comprehensive mental health action plan, 2013–2020 to achieve equity in leadership, governance, implementation of strategies, evidence and research (comprehensive mental health action plan 2013–2020; 2013). An extension to this plan, is the WHO comprehensive mental health action plan of 2013–2030, advocating for improved leadership in mental health and for action to be immediate to improve upon mental health needs, worldwide. Objective 1 of the mental health action plan specifically relates specifically to strengthening effective leadership in mental health which is directly supports the advance of lived experience value and leadership in mental health (World Health Organization, 2021).

### The World Health Organization report, 2022

The current WHO report places emphasis on the role that we all play in helping transform mental health for all. It calls for action from stakeholders to invest in mental health and include people with lived experience in engagement as they are vital in not only improving mental health systems but also improving services and outcomes (World Health Organization, 2022).

The report further confirms that people with lived experience are important agents of change in improving awareness on mental health, therefore, also able to reduce stigma, and they are in a better position to inform stakeholders of what is working and what is not working within and across all sectors (World Health Organization, 2022).

### The Lancet Commission on ending stigma and discrimination in mental health, 2022

The Commission's report is a result of a culmination of evidence and efforts of co-production and efforts of lived experience and their voices placing emphasis on how stigma negatively influences potential help seeking and that stigma is often worse than the condition itself. The Commission calls for action to "stop stigma and start inclusion" and that all stakeholders are responsible to action this call. The report asks governments, international agencies, employers, civil societies, healthcare providers and people with lived experience to work on ending stigma and it lays out eight recommendations broken down for each stakeholder (Thornicroft et al., 2022).

The key finding from the report confirms that people with lived experience are critical in advocating for a elimination of stigma through leading or co-leading positions and interventions in anti-stigma reduction programmes (Thornicroft et al., 2022). By involving lived experience in leading and co-leading programmes, we promote an increase in autonomy and resilience, enhance

empowerment and support for people with lived experience to turn their journeys of a mental health condition into a positive good for the public. People with lived experience bring unique expertise in the work they do regardless of the capacity they hold when they take on roles of consultants, members on advisory boards, committees and councils and so forth. They have been recognised as important players in service delivery, development and leadership (Sunkel and Sartor, 2022).

### Shared decision-making on the regional, national and international level

People with lived experience can play different roles in mental health and service provision and delivery, taking on roles such as advocates, peer-support workers, consultants, peer reviewers, researchers and lecturers. They have the ability to give back to the community, in an individual capacity and in collaboration with other professionals, on a national, regional and international level of decision-making.

As lived experience are active role players and stakeholders in engagement on mental health matters, no decision-making or changes to mental health policies and frameworks can be made without first consulting them for their expertise. The rationale behind this is that decisions cannot be made on behalf of persons directly affected by decisions, without first having sought their input. Excluding lived experience in itself goes against the person-centric, wholeness, human rights approach that promotes positive outputs and engagement.

### A collaborative model of best practice

In a review article by Aoki et al. (2022), shared decision-making was investigated in depth between clinicians and patients but the message therein is that a shared decision making approach is, in fact, beneficial and it creates a partnership approach where we are moving away from decisions made purely by clinicians but instead also engaging with patients and service users in the process. The authors furthermore emphasised that "shared decision-making is an ethical imperative and has been gaining support as a key principle of the delivery of person-centred care" (Aoki et al., 2022).

Engagement with lived experience must be based on the premise of enhancing diversity, respect, equality and equity and stakeholders ought to be better equipped within their own organisations and structures on how to meaningfully and authentically engage people with lived experience. Participation includes full empowerment and involvement in mental health advocacy, policy, planning, legislation, programme design, service provision, monitoring, research and evaluation" (World Health Organization, 2022).

### Guiding principles for the involvement of lived experience in decision-making

- *Mutual respect and trust*: Contributing to a space wherein participants feel respected and trusted for their competency, skills and reliability.
- *Transparency*: Ensuring participants have access to all relevant information pertaining the project and that there is clarity on the type of engagement and the process.
- *Non-discrimination*: Treating participants in the same way, free from judgement complimenting equal and fair treatment.

Including lived experience expertise from the onset of the project until the completion of project.

- *Non-tokenism*: Creating an approach that ensures diversity among participants without following approaches contributing to power imbalance.
- *Reasonable accommodation*: Ensuring flexibility in conditions of engagement in the sense of being adaptable to make reasonable changes during the engagement process to help participants with requests for accommodation to effectively perform their roles.
- *Flexibility*: Ensuring stakeholder flexibility in the working conditions of the engagement process, such as allowing for unexpected requests from participants particularly if related to their mental health. This will require an adaptable framework for increased productivity and engagement.
- *Diversity and equality*: Ensuring that different population groups are involved, considering age, race, sexual orientation, gender, religion and so forth without distinction.
- *Empowerment*: Ensuring participants are empowered by the process itself; ultimately valuing and recognising their participation and insights and to also ensure that they participate from the pre-project phase, and throughout until the end of the project.
- *Clear communication*: From the onset, setting transparent instruction on what is required from participants and clearly defined turnaround times and deadlines for completion of project tasks.

Implementing strategies that align with the effective engagement principles will promote a meaningful collaborative model of practice, reducing the practice of only including lived experience expertise midway in an engagement process. Acknowledging and appreciating lived experience expertise and contribution, creates a safe space for engagement, open dialogue and enhances opportunities for ongoing collaboration. Information and input provided by people with lived experience during the engagement process, must be included in the project summary and recommendations (if applicable), failing which, rendering the engagement unauthentic, ineffective and not meaningful.

## Lived experience monetary value: remunerating lived experience expertise

Acknowledgement and appreciation of lived experience expertise is not limited to verbal cues, it is essential to value the time and expertise of participants by providing remuneration for their time and participants as one would do for other professionals in the field.

Now, that lived experience expertise has received global recognition and acknowledgement from key stakeholders, it is imperative to bring to attention the limited progress made in terms of compensating lived experience for their expert participation and contribution to mental health related work. There needs to be a renewed emphasis and drive towards a global shift in this outdated yet current mindset. Stakeholders, local and global partners need to create a culture that recognise people with lived experience as experts in their own right, and not as volunteers.

Experts by experience contribute a level of expertise that cannot be found in any other profession. Besides lived experience expertise, people with lived experience often contribute additional expertise from academic qualification or non-academic skills that are not directly related to a mental health profession. Diverse lived experience expertise further bring knowledge and perspectives of different social and economic contexts.

## Lived experience: in peer-support work

In service delivery, people with lived experience play an important role as informal or formal peer-support workers and in a healthcare setting, peer-support workers have been known to enhance access to and quality of mental health care services. Peer support is a valuable resource as this form of service delivery enables an environment that is safe, free from judgement and sets the stage for recovery to take place. This role is unique as not only can peer support be used in the context of mental health services but is particularly useful in lived experience engagement and consultation sessions where discussions around personal experiences may become triggered and such a person may benefit from having a peer supporter available.

Lived experience is spearheading the future in peer-support work as confirmed by Basset et al. (2010) in a co-authored article titled lived experience leading the way.

Their article provides us with the 12 principles of peer support which needs much needed acknowledgement. These principles are mutuality, solidarity, synergy, sharing with safety and trust, companionship, hopefulness, focus on strengths and potential, equality and empowerment, being yourself, independence, reduction of stigma, respect and inclusiveness. Therefore, all the roles that people with lived experience take on, whether as service providers or as advisors, should be valued and remunerated to avoid potential consequences of dis-empowerment, exploitation and recreation of the "us and them" phenomenon.

## International key human rights instruments as applied to mental health

Legal frameworks exist to address breaches and violations in human rights that affect the livelihood of people living with mental health conditions. The premise is that everyone is equal before the law and deserves equal treatment and protection, despite this, human rights violations continue to be a major concern in health and mental health. While there are continued advocacy efforts to make change and create an ideal world free of violations, there is still much work to be done, globally and inclusively with all relevant stakeholders including persons with lived experience. Violations are often a result of the stigma, myths and misconceptions associated with mental health conditions (Arena Ventura, 2014).

The Office of the High Commissioner for Human Rights (OHCHR) is the leading United Nations (UN) entity in the field of human rights, and its premise is to ensure the promotion and protection of human rights for everyone.

## The Universal Declaration of Human Rights

Articles 1 and 2 of the Universal Declaration of Human Rights (United Nations General Assembly, 1948) states that all human beings are born free and equal with dignity and that these universal, inalienable rights exist without distinction of any kind. Notice the word used to describe human rights is inalienable; meaning that human rights are not provided or given to people by states or governments but are rights inherited at birth.

International key instruments provide the foundational legal framework for international human rights to which we are all bound. It is therefore encouraged that all persons become familiar with these documents and know their rights. Law is a powerful tool for protection but it is often overlooked and not referred to by

service users for fear of further stigmatisation and lack of resources in country (Gostin and Gable, 2004).

## The International Covenant on Economic, Social and Cultural Rights (1966) (ICESCR)

The ICESCR is a central instrument that deserves attention and acknowledgement for its promotion of equal pay for equal work. Article 7 of the covenant confirms and recognises that everyone has the right to enjoy favourable working conditions which ensure remuneration is provided to all workers (and that would include persons with lived experience who work as consultants, service providers and employees). Article 7 subsection (i) thereof also states that wages must be fair and that equal remuneration be given for equal work, without any distinction in kind. At the end of the day, persons with lived experience should be living for themselves and their families; they too have other responsibilities to fulfil.

## The Convention on the Rights of Persons with Psychosocial Disabilities (CRPD)

Article 27 of the Convention on the Rights of Persons with Psychosocial Disabilities (CRPD) (United Nations, 2006), on work and employment, promotes and protects the key principle for the remuneration of people with disabilities, including people with lived experience of a mental health condition. It promotes accessibility to the labour force and the opportunity to have a living and to be in a work environment that is both open and inclusive to persons with disabilities.

The CRPD echoes the rights of persons with disabilities as a milestone in human rights protection, offering people with psychosocial disabilities the opportunity to hold their governments accountable for the realisation of their rights. What this means is that Member State parties to the Convention and to the International Covenant on Economic, Social and Cultural Rights (United Nations [General Assembly], 1966), can be held accountable for the realisation of these rights.

## Equity and non-discrimination

A rights-based approach to health is essential and requires that health policies and programmes prioritise the needs of individuals towards greater equity, a principle that has been echoed in the 2030 Agenda for Sustainable Development and Universal Health Coverage (United Nations, 2015) and reaffirmed in 2019. Thereby pledging to "leave no one behind" as promoted by WHO. Responding to the needs of people, promotes respect, trust and shows appreciation for work done, the recipient will inevitably feel empowered, useful and proud of their meaningful contribution.

## Equal pay for equal work

Stakeholders including persons with lived experience must return to and refer back to available international law to protect their rights. For example, by referring to the International Covenant on Economic, Social, and Cultural Rights one could spark conversation on remuneration for lived experience expertise on the basis of "equal pay for equal work" as per law. From this conversation, parties involved in the work process may generate agreed upon solutions to issues that are prevalent and creating barriers to meaningful collaboration.

## Voice of lived experience

The Global Mental Health Peer Network (GMHPN), a leading international peer led lived experience organisation has taken a big leap in advocating for equal remuneration for lived experience expertise and participation. The GMHPN's work is focused on capacity building, empowerment and development of global lived experience leaders.

More than a few members of GMHPN's country leadership committee, from all over the world have generously offered their insights into how the felt the first time they were compensation for work done in their mental health advocacy work.

> Being paid fostered my empowerment, autonomy, self-confidence and dignity and was a recognition of the genuine expertise gained over years of struggle with mental health issues – Arnaud Poitevin, France.

> I felt as if I was being recognized for the input I can give as well as someone with first-hand experience. I also felt it was the beginning of a career that I actually enjoy and that it is going to open up many doors for me – Anonymous, Zimbabwe.

Other GMHPN members shared their perspectives on what they believe is most important for stakeholders to consider when including lived experience in projects.

> I was valued, it's a recognition because the lived experience knowledge is priceless, we had been through valuable lessons, that worth more than anything. When I decided to contribute my expertise, my intention is helping others. So, it's about respect and dignity – Agus Sugianto, Indonesia.

> "As lived experience people should be harbingers of hope and trusted to be sincere in their empathy and examples of living in positive growth, they should be able to demonstrate their awareness of their own limits and boundaries, their newly balanced life which would include a solid self-care plan, and examples of co-regulation while living with their experiences and last but not least their connectivity to those who can help support those in need" – Sylvio Gravel, Canada.

> "Our world is becoming increasingly culturally diverse and it's really important to factor in the socio-cultural backgrounds of people with lived experience while engaging them in projects. Society and culture massively impact people's lives and well-being, deeply intertwining with—their worldview, decision-making abilities, interactions with others, language, identity, and the social and cultural stigmas that often come attached—all of which can enable a person to propel a project forward, or hold them back in their potential" – Muskan Lamba, India.

## Recommendation

Based on existing international laws and the perspectives of people with lived experience (as per above), this article would like to share its recommendations to improve upon implementation and engagement by all parties so that that equity in processes becomes global good practice.

A call to action to all local and global stakeholders in the global mental health field to initiate the implementation of the effective engagement principles into their projects and partnerships with people lived experience; ensuring a truly collaborative model of good practice. Thereby, meeting the principle of meaningful and authentic engagement which in turn leads to empowerment, support, fairness and equality; an approach that will hopefully benefit all parties involved and aid in achieving common goals of positive change in mental health work.

## Conclusion

Despite the established consensus on the experiential value of the expertise of people living with mental health conditions, there remains a gap in acknowledging their monetary value in the global mental health sector in service delivery and co-production of projects and programmes. Therefore, reducing opportunities for lived experience advocates to drive systemic change and make recommendations for positive change. Appreciating the current evidence on international law and human rights instruments, people with lived experience and peer-led organisations continue their advocacy efforts for equal and fair treatment of persons with disabilities including psychosocial disabilities for reasonable, fair and equal remuneration for work done. Efforts among stakeholders must improve in order to work within a collaborative model of practice. To do this, one needs to ensure the guiding principles of engagement with lived experience such as transparency, mutual trust and respect, collaboration, non-tokenism and non-discrimination are implemented during the engagement process thereby promoting healthy working relationships and equal treatment among partners.

In concluding, with proper planning, prioritisation, practices of equal treatment and revised budgetary limitations, positive change and transformation can be reasonably expected. We remain hopeful that the current gaps will be bridged through collaborative efforts and implementing a meaningful and authentic collaborative model, worldwide.

**Open peer review.** To view the open peer review materials for this article, please visit http://doi.org/10.1017/gmh.2023.24.

**Financial support.** This research received no specific grant from any funding agency, commercial or not-for-profit sectors.

**Competing interest.** The author declares no competing interests exist.

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
