## [Reviewer Report]

Thank you very much for the opportunity to review this article, which brings an important perspective on a much needed subject - the remuneration of people with lived experience for their work. People with lived experience see the profound meaning in their participation in decision-making concerning the systematic development of mental health care systems. However, very often they are not paid or are paid insufficiently for their important contribution.

Principle “equal pay for equal work” mentioned in this article should be seen as the main principle of the remuneration of lived experience, based on dignity of and respect to the people with lived experience. The lived experience of mental health problems constitutes non-transferable knowledge and skills and brings a very important perspective on systems of mental health care. The expertise of people with lived experience is essential for decision-making in the field of mental health and, as such, should be remunerated.

I appreciate the effort of the author to highlight the necessity of the fair remuneration of an expertise of the lived experience. I consider this article as a great contribution to the global mental health debate concerning the effective involvement of people with lived experience in changes of mental health care systems. I am convinced that without expertise of people with lived experience systematic and effective changes of mental health care systems are impossible.

PWLE have the right to be paid for their work fairly. The fair remuneration increases the empowerment of PWLE on the individual level and brings them the feeling of dignity and usefulness.

However, I think the article could benefit from including some of my following suggestions:

1. Minor restructuring

a. From my point of view some headlines could be formulated more clearly; for example the headline: “Lived experience value: experiential” could be replaced with “International resources supporting lived experience value”; instead of the headline “List of considerations as well as guiding principles for stakeholders to refer to when working and engaging with people with lived experience” the headline “Guiding principles for the involvement of lived experience in decision-making” could be used; the headline “What are we saying with all this” could be replaced with “Equal pay for equal work” in order to put attention to this principle. Maybe instead of the headline “Perspective from global lived experience…..” the headline “Voice of lived experience” could be used.

b. In order to ensure the more understandable structure of this article I recommend to add some new headlines: the first one before the paragraph beginning with “In a review paper by Aoki….”, the second before the paragraph beginning with “Engagement with lived experience……..”, the third before the paragraph beginning with “Implementing strategies that align with the effective engagement….” and the fourth headline before the paragraph beginning with the sentence “Articles 1 (one) and 2 (two)……”. Maybe the concrete headlines could be: “Shared decision-making on the individual level”, “Shared decision-making on the regional, national and international level”, “Collaborative model of practice”, “The Universal Declaration of Human Rights” or something like that.

c. It would be great if the headline “Lived experience Monetary value: Remunerating lived experience” was moved before the paragraph beginning with the sentence “Acknowledgment and appreciation of lived experience…..” and the headline “Recommendation” before the paragraph beginning with the sentence "Based on existing international laws…..”

d. From my point of view, it would be clearer if the paragraph beginning with the sentence “Legal frameworks exist……” was moved at the beginning of the section titled “International Key Human Rights instruments as applied to mental health”.

2. I consider the document of WHO “Comprehensive mental health action plan 2013-2030” namely its Objective 1 as very important material supporting the lived experience value, so it would be great to mention it explicitly in the section titled “Lived experience value: experiential”.

3. Concerning the CRPD, its Article 27 (Work and employment) brings the key principle for remuneration of people with disability, it would be great to mention this article explicitly.

4. For better understanding of the section titled “Shared decision-making: a good practice model” I recommend starting this section with explanation that PWLE can play different roles (peer support workers, peer advocates, peer researchers, peer lecturers etc.), when they are involved in decision-making and that they cooperate with other professionals on the individual, regional, national and international level of decision-making.

Thank you very much for the possibility to make this open peer review. As woman with lived experience I fully understand the problematics described in this article and I very recommend to publish it in your journal.

---

## [Reviewer Report]

This paper makes the case for the value of lived and living experience leadership and partnership in mental health reform efforts. The paper rightly calls for those in lived experience leadership roles or who are bringing their lived experience to mental health reform efforts or to service delivery as peer workers to be better remunerated for their efforts. This call is grounded in human rights principles and in key international human rights instruments.

The author cites two recent reports, including the Lancet Commission on Ending Stigma and Discrimination and a June 2022 report from the WHO, as evidence of commitment of the value of lived experience leadership in global mental health reform efforts. It could be helpful for the author to reflect that these commitments are not new and build on decades of advocacy from people with lived experience, their families, friends and support people to convince those in the global mental health community and those leading reform efforts in their own jurisdictions of the value of lived experience leadership.

There are a number of places in the article where the author might find it useful to reflect critically on the concepts being discussed and whether there might be an approach which positions lived experience even more strongly.

For example, the author suggests “Shared Decision Making” as an example of moving towards a more person centred mental health system. While “Shared Decision Making” is certainly more desirable than decisions being made entirely by clinicians, a more human rights centred approach would be to move to "’Supported Decision Making" as described in the Convention on the Rights of Persons with Disability. This approach puts decision making in the hands of those with lived experience and encourages clinicians and other support people to provide information and others supports so that the person is able to make decisions for themselves. This approach embodies agency and autonomy and is more aligned with the human rights principles the author has grounded the work in.

Another example, is in reference to the concept of equality. The author might find it useful to reflect more critically on the principle of non-discrimination and whether treating participants ‘equally’ is the goal, or whether there might be value in considering the concept of “equity” and asking questions of who has the opportunity to participate, is this inclusive of a diverse range of experiences - including those who have experienced harm by the system or who face additional barriers to participation, what support they might need to participate and how the system needs to change in order to welcome and value their participation.

Prior to publication, the article could benefit from some proof reading to avoid unclear or repetitive phrasing eg “how to correctly engage people with lived experience in engagement” might be rephrased as ‘how to engage people with lived experience meaningfully.’

Overall, this article provides an important contribution to making the case for valuing lived experience leadership as part of global mental health reform efforts. Thank you for the opportunity to review the article and provide some suggestions to strengthen it.

---

## [Reviewer Report]

Thank you very much for the opportunity to review this article, which brings an important perspective on a much-needed subject - the remuneration of people with lived experience (PWLE) for their work, again.

I recommend this article for publishing. My original suggestions I recommended in my first review were largely accepted. I consider this article clearer and more readable now.

The revisions used supported the author´s deep insight into the issue of compensating people with lived experience for their work in the field of developing mental health care systems in the world.

The addition of important documents amplified the emphasis on the right of PWLE to be rewarded based on the principle “equal pay for equal work”.

However, I would like to point out two small inaccuracies.

I would like to mention that The UN Convention on the Rights of Person with Disabilities (CRPD) was adopted in December 2006 and entered into force in 2008. From my point of view, it would be great to cite both data to avoid any doubts.

The correct title of CRPD is “The United Nations Convention on the Rights of Persons with Disabilities”. The author once used the title: “The Convention on the Rights of Persons with Psychosocial Disabilities”. It would be great to exclude the word “psychosocial” from the headline of relevant paragraph.